# Position Is All You Need: A Free Lunch Token Compression Strategy for MLLM-based Referring Expression Segmentation

Yuhan Liu [* 1]   Yixiong Zou [* 1]   Yuhua Li [1]   Ruixuan Li [1]

## Abstract

Referring Expression Segmentation (RES) aims to generate pixel-wise segmentation masks from complex and implicit textual queries. While recent advances in Multimodal Large Language Models (MLLMs) have substantially boosted RES performance, their prohibitive computational overhead remains a critical bottleneck, which, however, is rarely explored. To fill this gap, we first evaluate typical token compression methods on this task and observe a surprising performance degradation. In this paper, we aim to understand this phenomenon for a solution. By extensive experiments, we find that token compression for RES requires preserving the original position embeddings and local neighboring spatial structures, indicating that visual token position information is far more critical than in other tasks. Building on this insight, we ask: *Can we design the token compression method **purely based on the position information**?* Therefore, we propose PAYN, a plug-and-play, training-free token compression method that relies solely on position information. PAYN retains tokens that are adequately distributed in every local neighboring region while strictly preserving original positional indices, thereby maintaining spatial relational consistency. Experiments on multiple RES benchmarks demonstrate that our method outperforms existing token compression methods, verifying that position is indeed all you need for token compression in the MLLM-based RES task. Codes are avaliable at https://github.com/YuhanLiu231/PAYN.

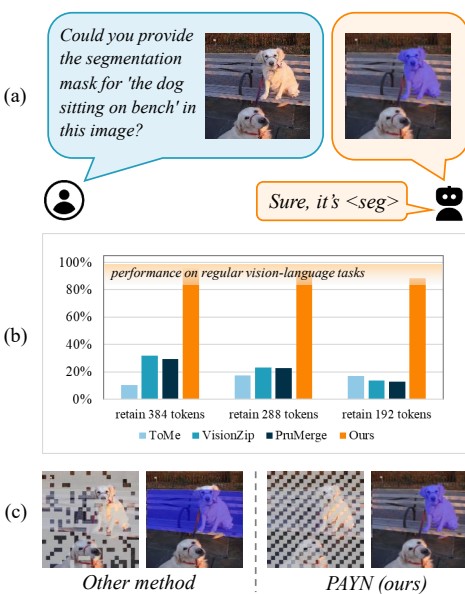

*Figure 1.* (a) The Referring Expression Segmentation (RES) task. (b) While representative token compression methods can preserve over 90% of the original performance on regular vision-language tasks, they suffer from severe performance degradation on the RES task, retaining barely around 20%, which we aim to understand and handle in this paper. (c) Visualization of the retained tokens and segmentation results of our method and competing method. Our free-lunch method, which relies solely on positional information, achieves superior performance.

## 1. Introduction

The Referring Expression Segmentation (RES) task predicts pixel-level segmentation masks given complex or implicit textual descriptions (Hu et al., 2016; Ding et al., 2025). As shown in Fig. 1(a), unlike traditional category-based segmentation, which uses labels like "dog", RES allows free-form expressions such as "the dog sitting on bench" for object descriptions, offering flexible and user-friendly segmentation. To handle this challenging task, current works (Wang et al., 2023; Lai et al., 2024; Xia et al., 2024; Pi et al., 2024; Ren et al., 2024; Rasheed et al., 2024; Lan et al., 2025; Wei et al., 2025) leverage Multimodal Large Language Models (MLLMs) (Bai et al., 2023; Liu et al., 2024; Lu et al., 2024; Chen et al., 2024c) for the understanding of complex text

---
[*]Equal contribution   [1]School of Computer Science and Technology, Huazhong University of Science and Technology, Wuhan, China. Correspondence to: Yixiong Zou <yixiongz@hust.edu.cn>, Yuhua Li <idcliyuhua@hust.edu.cn>.

*Proceedings of the 43$^{rd}$ International Conference on Machine Learning*, Seoul, South Korea. PMLR 306, 2026. Copyright 2026 by the author(s).

and visual inputs, focusing on designing effective multimodal representation formats for segmentation decoding. However, due to the quadratic scaling of attention computation with input sequence length, MLLMs suffer from high inference cost (Chen et al., 2024a; Lin et al., 2025; Yang et al., 2025b), which limits the practical deployment of RES. To the best of our knowledge, few works have studied the acceleration for the MLLM-based RES task.

To fill up this underexplored area, we begin with token compression methods as a widely-adopted acceleration strategy (Liang et al., 2022; Bolya et al., 2023) for MLLMs, which typically reduces the inference cost by removing or merging substantial redundant visual tokens (Wang et al., 2025; Yang et al., 2025a; Chen et al., 2026; Zhang et al., 2026; Tong et al., 2026). However, by applying prevailing token compression methods to MLLM-based RES models, we surprisingly find that these methods lead to severe performance degradation, as shown in Fig. 1(b). When retaining the same portion of visual tokens, these methods can maintain over 90% of the original performance on regular vision-language tasks, but can hardly maintain even about 20% of the original performance on the RES task, leading to almost collapsed results after the token compression.

In this paper, we aim to understand why this problem occurs, and specifically design token compression methods for the MLLM-based RES task based on this interpretation. We first take a detailed evaluation of current token compression methods on the RES task. By (coarsely) dividing current methods into attention/similarity-based ones (Bolya et al., 2023; Yang et al., 2025b; Shang et al., 2025) and diversity-based ones (Wen et al., 2025; Alvar et al., 2025; Zhang et al., 2025a), we find that the diversity-based ones show much better performance, where the core reason is the maintenance of original position embeddings, as widely adopted in diversity-based methods. To further understand why position embeddings are much more important in the RES task than in other tasks, through extensive experiments, we find that it is because the RES task requires the precise prediction of **every visual token**'s label, while other tasks typically only require the understanding of the **semantic part of visual tokens**, which makes a small perturbation of local neighbor positions (e.g., not maintaining original position embeddings) influences much more on the RES task than on other tasks. This leads to our insight for the token compression methods in MLLM-based RES task: **position information of visual tokens is much more important than in other tasks**.

Based on this insight, now that the position information is much more important, *can we design the token compression method **purely based on the position information***? Therefore, we propose PAYN, a plug-and-play, training-free token compression method that relies solely on positional

information without considering any semantic content, effectively serving as a free-lunch solution, as illustrated in Fig. 1(c). Specifically, our method strictly preserves the original positional indices while aiming to retain tokens that are spatially averagely distributed, thus adequately preserving local neighboring spatial structures. We provide two instantiations of this approach: checkerboard-style spatial sampling and farthest point sampling. Extensive experiments on multiple RES benchmarks demonstrate that our method outperforms existing state-of-the-art token pruning approaches, verifying that **position is all you need for token compression in the MLLM-based RES task**.

In summary, our key contributions are as follows:

- We find that existing token compression methods suffer from severe performance degradation in the MLLM-based RES task, which is underexplored.

- Through experiments and analysis, we observe that positional information of visual tokens plays a substantially more critical role in RES than in other tasks.

- Guided by this observation, we propose PAYN, a simple but effective free-lunch token compression method that relies solely on positional information.

- Extensive experiments demonstrate that our method outperforms existing SOTA token pruning approaches, validating that position is all you need for token compression in the MLLM-based RES task.

## 2. Related Work

### 2.1. MLLM-Based Referring Expression Segmentation

Referring Expression Segmentation (RES) aims to segment the pixels of a target object in an image based on a natural language referring expression (Lai et al., 2024; Zhang et al., 2024; Chen et al., 2024b; Wu et al., 2024; Qian et al., 2025; Zhu et al., 2025a; Liu et al., 2026). With the rapid development of multimodal large language models (MLLMs) (Liu et al., 2024) and powerful segmentation models like SAM (Kirillov et al., 2023), significant progress has been made in RES. LISA (Lai et al., 2024) pioneers the embedding-as-mask paradigm by extending the MLLM vocabulary with a special <seg> token to guide a segmentation decoder. Following this paradigm, several subsequent works extend MLLM-based RES from different perspectives. GSVA (Xia et al., 2024) employs multiple <seg> tokens along with a <rej> token to handle multiple objects and reject null targets. PixelLM (Ren et al., 2024) replace SAM with a lightweight pixel decoder and introduce a learnable segmentation codebook for mask generation. InstructSeg (Wei et al., 2025) provides a unified framework for performing language-instructed segmentation across both images

and videos. In contrast to these embedding-as-mask approaches, Text4Seg (Lan et al., 2025) proposes a text-as-mask paradigm that reformulates image segmentation as a text generation problem by encoding images into sequences of semantic textual descriptors. Another line of work (Wang et al., 2023; Peng et al., 2024) leverage MLLMs to produce polygon coordinates for mask prediction, which often struggle to achieve satisfactory performance.

### 2.2. Visual Token Compression

In MLLMs, visual signals exhibit much higher spatial redundancy than text, motivating visual token compression to improve efficiency. From the perspective of optimization requirements, some methods require finetuning the model (Cai et al., 2025; Ye et al., 2025b), some determine pruning strategies using a calibration dataset (Lin et al., 2025; Ye et al., 2025a), while others operate in a training-free manner (Yang et al., 2025b; Zhang et al., 2025b). From the perspective of compression mechanisms, existing methods can be broadly categorized into similarity-based approaches, e.g., ToMe (Bolya et al., 2023), diversity-based approaches, e.g., Dart (Wen et al., 2025), DivPrune (Alvar et al., 2025), and attention-based approaches, e.g., VisionZip (Yang et al., 2025b), PruMerge (Shang et al., 2025), Vispruner (Zhang et al., 2025a). Similarity-based methods merge highly similar tokens into fewer representative tokens, for example, ToMe implements this using bipartite matching. Diversity-based methods also exploit token similarity, but instead aim to retain tokens that are maximally different from each other. For instance, DivPrune formulates token compression as a Max-Min diversity problem, selecting a subset of tokens with the largest internal differences. Attention-based methods retain high-attention tokens while pruning low-attention ones, and can be combined with similarity-or diversity-based methods. While these methods perform well on sparse prediction tasks, visual token compression for dense prediction tasks such as RES is largely underexplored (Kong et al., 2025). In this work, we propose a training-free, plug-and-play token compression method tailored for RES.

## 3. Rethinking Token Compression for Referring Expression Segmentation

### 3.1. Preliminary

MLLM-based Referring Expression Segmentation (RES) aims to generate a pixel-level segmentation mask $\mathcal{M}$ for a target object in an image $I$, conditioned on a natural-language referring expression $T$. The input image $I$ is encoded by a vision encoder (e.g., SigLIP (Zhai et al., 2023)) into a sequence of visual tokens $\mathbf{X}_v \in \mathbb{R}^{M \times d}$, while the referring expression $T$ is encoded into a sequence of language tokens $\mathbf{X}_t \in \mathbb{R}^{N \times d}$. Typically, the number of visual tokens

$M$ is much larger than the number of text tokens $N$, leading to substantial computational overhead when processed by the Large Language Model (LLM). To this end, a compression operator $\mathcal{F}_{comp}$ is employed to project the raw visual sequence into a compact representation $\mathbf{C}$:

$$\mathbf{C} = \mathcal{F}_{comp}(\mathbf{X}_v), \quad \text{s.t. } |\mathbf{C}| < |\mathbf{X}_v| \qquad (1)$$

Here $\mathbf{C} = \{c^k\}_{k=1}^K$ denotes a set of $K$ compressed visual tokens. These tokens are concatenated with the language tokens $\mathbf{X}_t$ and fed into LLM:

$$\mathbf{H}_{out} = \text{LLM}(\text{Concat}(\mathbf{X}_t, \mathbf{C})). \qquad (2)$$

$\mathbf{H}_{out}$ denotes the high-level guidance for the segmentation task, which may take the form of $<\text{seg}>$ embeddings, textual representations, or other structured signals. Finally, the segmentation mask $\mathcal{M}$ is derived through a decoder (e.g. SAM) conditioned on both the original image features $f_{img}$ and the guidance $\mathbf{H}_{out}$:

$$\mathcal{M} = \text{Decoder}(f_{img}, \mathbf{H}_{out}) \qquad (3)$$

### 3.2. Evaluation of Existing Token Compression Methods

In prior experiments, we observed that several existing token compression methods suffer from significant performance degradation on the RES task. To delve deeper into this phenomenon, we conduct a more systematic evaluation of a broader range of token compression strategies. All experiments are conducted on a unified RES baseline, namely Text4Seg (Lan et al., 2025), with LLaVA-1.5-7B (Liu et al., 2024) as the backbone, where the number of visual tokens is reduced from 576 to 192. From the perspective of their underlying mechanisms, these methods can be broadly divided into four categories: (1) similarity-based (Bolya et al., 2023) (2) attention & similarity-based (Yang et al., 2025b; Shang et al., 2025) (3) diversity-based (Wen et al., 2025; Alvar et al., 2025) (4) attention & diversity-based (Zhang et al., 2025a). The results are shown in Table 1(top).

In table 1(top), we observe that **(attention&) diversity-based methods significantly outperform (attention&) similarity-based ones**. Besides, we also observe that (attention&) similarity-based methods often employ token merging or clustering, which reorganizes tokens into a contiguous sequence with re-indexed positions (e.g., 0, 1, 2), but (attention&) diversity-based methods typically perform token pruning, which preserves the original position indices of the retained tokens (Fig. 2). Therefore, we hypothesize two potential reasons for this disparity. (1) (Attention&) diversity-based methods are inherently more suitable for the RES task. (2) The two types of methods differ in how positional information is handled after token compression.

To verify which of the two hypotheses holds, we conduct a set of controlled experiments. We isolate the attention-based

*Table 1.* Performance comparison of visual token compression methods on the RES task. (Top) Results of existing methods, where S, A, and D denote similarity-based, attention-based, and diversity-based token selection criteria respectively. (Middle) Ablation on A & S methods by removing S and evaluating the effect of position ids, where preserving position ids yields significantly better performance. (Bottom) Performance drops when removing position ids from D methods further supporting this observation.

| Method | Category | refCOCO | refCOCO+ | refCOCOg | Avg. |
|---|---|---|---|---|---|
| vanilla | - | 78.3 | 71.9 | 73.0 | 74.9 |
| ToMe | S | 13.8 | 12.0 | 12.8 | 12.9 |
| VisionZip | A & S | 11.6 | 10.1 | 8.7 | 10.3 |
| PruMerge | A & S | 10.2 | 9.8 | 8.5 | 9.6 |
| Dart | D | 64.0 | 57.6 | 60.1 | 60.6 |
| DivPrune | D | 67.3 | 60.5 | 62.6 | 63.6 |
| Vispruner | A & D | 65.7 | 60.9 | 61.1 | 62.8 |
| VisionZip | A | 10.6 | 9.2 | 8.4 | 9.5 |
| PruMerge | A | 10.3 | 9.0 | 8.6 | 9.4 |
| VisionZip w/pos id | A | 56.5 | 51.8 | 53.5 | 54.0 |
| PruMerge w/pos id | A | 56.3 | 52.7 | 53.4 | 54.2 |
| Dart w/o pos id | D | 10.3 | 8.9 | 9.2 | 9.5 |
| DivPrune w/o pos id | D | 10.2 | 8.8 | 8.9 | 9.7 |
| VisPruner w/o pos id | A & D | 9.9 | 8.8 | 9.0 | 9.3 |

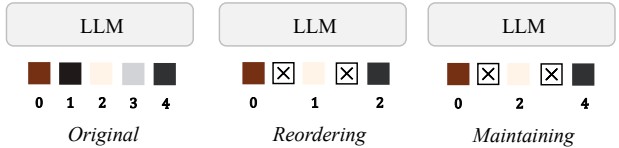

*Figure 2.* Different strategies in position indices of pruned tokens.

selection component, while keeping the number of retained tokens unchanged, with and without preserving the original position indices. The results are shown in Table 1(middle). We observe that preserving position indices yields a clear performance advantage and achieves results comparable to (attention &) diversity-based methods.

Furthermore, we remove the original position indices in (attention &) diversity-based methods and replace them with contiguous ones. As shown in Table 1(bottom), this leads to a sharp performance drop. Therefore, these results validate the second hypothesis, and we can conclude that:

> **Insight 1.** For the referring expression segmentation task, preserving the original positional embedding during token compression is essential.

### 3.3. The Role of Position in Sparse and Dense Prediction Tasks

Building on *Insight 1*, we also notice that (Lin et al., 2025; Ye et al., 2025b) mentioned that maintaining the original position indices leads to only around 2% difference in performance. This inspires us to further investigate why preserving the original positional embedding is crucial for RES, while having a much smaller impact on conventional vision-language tasks. We hypothesize that the discrepancy arises

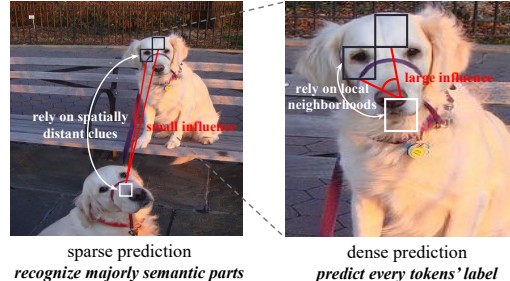

sparse prediction
*recognize majorly semantic parts*

dense prediction
*predict every tokens' label*

*Figure 3.* Different task requirements of sparse and dense prediction tasks lead to distinct sensitivities to relative positional shifts. The black boxes indicate simulated positional shifts.

from the different sensitivity of *dense* and *sparse* prediction tasks to local spatial consistency.

As shown in Fig. 3, sparse prediction tasks, such as Visual Question Answering (VQA), primarily rely on a few discriminative semantic patches, which can be spatially distant from each other. As a result, these tasks are less sensitive to precise relative positions, and therefore suffer less performance degradation when positional indices are altered after token compression. In contrast, dense prediction tasks like RES require consistent modeling of local neighborhoods to produce pixel-wise predictions. If the original positional indices are not preserved during token compression, tokens that are far apart in the image may be assigned adjacent indices, leading to severe distortion of local spatial relationships and consequent performance degradation.

To validate the above hypothesis, we design experiments by applying local spatial perturbations to the input images on both tasks. Specifically, we introduce two types of perturbations that primarily disrupt local spatial structures while largely preserving semantic information, as illustrated in Fig.4 (left). The first one employs Thin Plate Spline (TPS) interpolation (Wood, 2003) to induce smooth, continuous spatial deformations. The second divides image patches into local windows and randomly shuffles the patch order within each window, resulting in discrete local reordering. We evaluate the impact of these perturbations

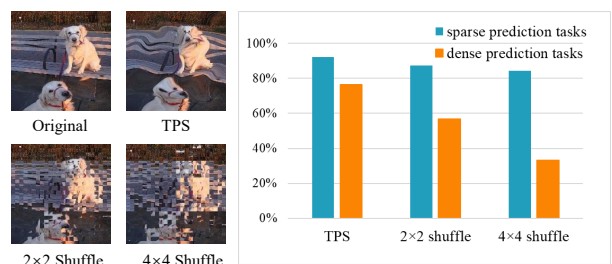

Original TPS

2×2 Shuffle 4×4 Shuffle

*Figure 4.* Sensitivity of dense and sparse prediction tasks to local positional perturbations. (Left) Two types of perturbations applied to input images, including continuous TPS-based deformations and discrete local patch shuffle. (Right) Dense prediction tasks suffer substantially larger performance drops than sparse prediction tasks under the same perturbation strength.

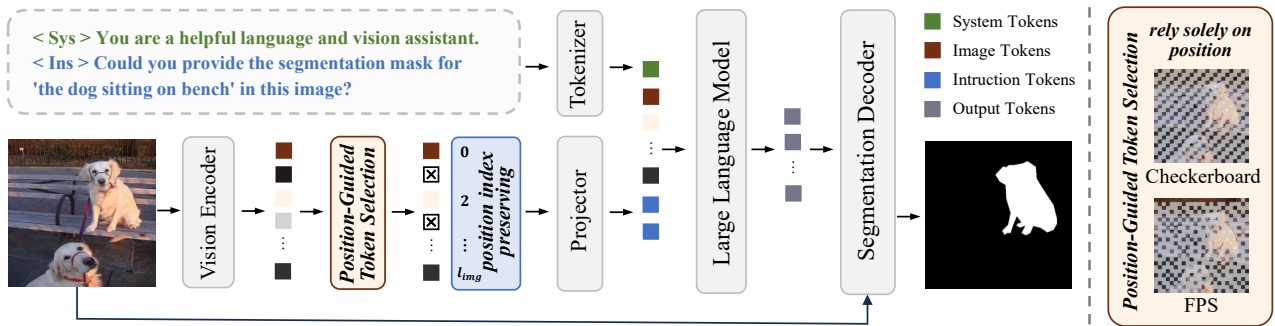

*Figure 5.* Our method consists of two components: (1) position-guided token selection, which retains tokens solely based on positional information to ensure adequate retained tokens in every local neighboring region, and is instantiated by two strategies, Checkerboard and FPS; and (2) a position index preserving strategy, which aims to preserve original positional information.

on representative sparse prediction tasks: MME (Fu et al., 2025), ScienceQA (Lu et al., 2022), and TextVQA (Singh et al., 2019), as well as dense prediction tasks across multiple RES datasets.

As shown in Fig.4 (right), **under the same local spatial perturbations, dense prediction tasks exhibit a substantially larger relative performance drop compared to sparse prediction tasks**. This observation indicates that dense prediction tasks are indeed more sensitive to local positional variations, thereby validating our hypothesis:

> **Insight 2.** Referring expression segmentation critically depends on the preservation of local spatial structures.

Building upon *Insights 1 and 2*, which highlight the strong dependence of RES on positional information, we naturally raise the following question: *Is position all you need for token compression in the MLLM-based RES task?*

## 4. Method

Guided by the two insights and the question posed above, we aim to investigate a token compression method for RES that **relies solely on positional information**, as shown in Fig.5. The approach is fully plug-and-play, training-free, and does not even require attention maps or feature information, effectively serving as a free-lunch solution.

### 4.1. Position-Guided Token Selection

**According to *Insight 2***, RES is particularly sensitive to local spatial variations and relies heavily on the preservation of local spatial structures. To ensure that all local regions are adequately represented after compression, we design to ensure enough maintained tokens in *every* local neighboring region. Therefore, we need to ensure broad and uniform spatial coverage of maintained tokens across the image, so we retain tokens that are maximally separated in the spatial positions. Below, we provide two instantiations of this

position-guided token selection strategy $\mathcal{F}_{comp}$.

**Checkerboard-Style Spatial Sampling**  We adopt a simple and efficient checkerboard-style spatial sampling strategy on a regular 2D grid of patch-level visual tokens indexed by spatial coordinates $(x, y)$. Given the target number of retained tokens $K$ and the number of total tokens $M$, the retained token set $C$ is defined as

$$\mathcal{C} = \begin{cases} \{(x,y) \mid (x+y) \bmod s = \phi\}, & K \leq M/2, \\ \{(x,y) \mid (x+y) \bmod s \neq \phi\}, & K > M/2, \end{cases} \quad (4)$$

where $\phi$ is a fixed offset (set to 0 in our implementation) that determines the checkerboard sampling pattern. The stride $s$ is defined as:

$$s = \left\lfloor \frac{M}{\min(K, \, M - K)} \right\rfloor. \quad (5)$$

The retained tokens preserve their original positional indices. This strategy results in a spatially uniform, checkerboard-like distribution of retained tokens, enabling broad coverage of local regions without relying on semantic information.

**Farthest Point Sampling (FPS).**  As a more flexible instantiation of position-guided token selection, especially when the number of retained tokens $K$ is very small, we perform farthest point sampling (FPS) directly in the spatial coordinate space. Starting from an initial randomly selected token, FPS iteratively selects the token that maximizes its minimum distance to the already selected set. Formally, at each iteration $t$, the next token is chosen as:

$$c^t = \arg \max_{c \in \mathbf{X}_v \setminus \mathbf{S}_{t-1}} \min_{c' \in \mathbf{S}_{t-1}} \|\mathbf{p}_c - \mathbf{p}_{c'}\|_2^2, \quad (6)$$

where $\mathbf{p}_c \in \mathbb{R}^2$ denotes the 2D spatial coordinates of token $c$. The procedure is repeated until $|\mathbf{S}_t| = K$, producing a set of well-dispersed tokens over the spatial domain, while retaining their original positional indices.

## 4.2. Position Index Preserving

According to *Insight 1*, preserving the original positional indices of retained image tokens is essential for RES, so we adopt a position index preserving strategy. Compressed visual tokens are first projected by a multimodal projector and then concatenated with text tokens to form the input sequence for the LLM:

$$\mathbf{X} = [\mathbf{X}_{sys}, \mathbf{X}_{img}, \mathbf{X}_{ins}, \mathbf{X}_{out}] \qquad (7)$$

where $\mathbf{X}_{sys}, \mathbf{X}_{img}, \mathbf{X}_{ins}, \mathbf{X}_{out}$ denote system prompts, image tokens, user instructions, and output tokens, respectively. Without position-preserving reindexing, positional indices are assigned monotonically from $0$ to $L-1$, where $L$ denotes the total sequence length. After compressing, we replace the positions of image tokens by the preserved original indices:

$$\mathbf{I}_{img}^{preserved} = \{i_1, i_2, \ldots, i_K\}, \quad K < |\mathbf{X}_{img}| \qquad (8)$$

while the positions of subsequent instruction and output tokens are shifted accordingly to maintain the overall sequence order. Formally, the reindexed positions are:

$$\mathbf{P}_{reindex} = [\mathbf{P}_{sys}, \mathbf{I}_{img}^{preserved}, \mathbf{P}_{ins}^{shifted}, \mathbf{P}_{out}^{shifted}] \qquad (9)$$

Position index preserving guarantees that the relative spatial structure of image tokens is retained in the LLM, enabling more precise dense predictions. After obtaining the LLM's output as high-level guidance, it is fed together with the image into a decoder to produce the segmentation mask.

## 5. Experiments

### 5.1. Experimental Setting

**Baselines and Models** We adopt two representative RES baselines: Text4Seg(Lan et al., 2025) and InstructSeg(Wei et al., 2025). Text4Seg follows the text-as-mask paradigm by encoding images into semantic textual descriptors, with LLaVA-7B as the MLLM backbone. In contrast, InstructSeg is a recent state-of-the-art model based on the embedding-as-mask paradigm, built upon Mipha-3B(Zhu et al., 2025b).

**Datasets and Metric** Following standard evaluation protocols (Lai et al., 2024), we evaluate our method on the RefCOCO, RefCOCO+ (Kazemzadeh et al., 2014), and RefCOCOg (Mao et al., 2016) benchmarks. These datasets include over 19,000 images for RefCOCO and RefCOCO+, and 26,711 images for RefCOCOg. RefCOCO allows expressions that reference both location and appearance attributes. RefCOCO+ focuses on appearance-based descriptions while limiting location cues. RefCOCOg contains longer and more complex expressions without restrictions on location references (Ding et al., 2025).

For evaluation, we adopt the cumulative Intersection-over-Union (cIoU) metric, calculated as the total intersection over the total union of all predicted masks.

## 5.2. Main Results

We compare our method with other training-free, calibration-free token compression methods (Bolya et al., 2023; Yang et al., 2025b; Wen et al., 2025; Zhang et al., 2025a; Alvar et al., 2025; Zhu et al., 2026). Unlike token compression in other tasks, whose baselines are plain MLLM models such as LLaVA, RES baselines vary in model structures and complexity. We therefore adopt different compression ratios for each RES baseline to balance segmentation performance and inference efficiency. Table 2 reports results on the Text4Seg baseline, shows that our PAYN achieves the best performance across nearly all benchmarks, with its average results substantially outperforming existing methods. The performance gap is especially noticeable when fewer tokens are retained. Following (Zhu et al., 2026), we further evaluate PAYN on InstructSeg. As illustrated in Table 3, our method continues to outperform alternative approaches.

### 5.3. Ablation Studies

**Validating the *Position Is All You Need* Perspective** The results in Table 2 and Table 3 demonstrate that our position-based approach outperforms attention-, similarity-, and diversity-based token compression approaches. To explore other factors that might influence token selection, we considered several alternative strategies:

(1) Random selection, serving as a baseline;
(2) Selecting tokens with larger L2 norms, assuming they carry more information;
(3) Selecting edge or contour tokens to preserve boundary information important for segmentation, using two different edge-detection operators: Sobel (Sobel et al., 1968) and Canny (Canny, 2009);
(4) Uniform sampling within clusters, where tokens are first clustered and then an equal number of tokens are randomly sampled from each cluster, either on the original image or on patch embeddings;
(5) Farthest point sampling in the feature space.

*Table 4.* Comparison of alternative token selection criteria such as semantic information or edge cues, the original positional indices of the selected tokens are preserved. The position-based strategy consistently outperforms all alternatives, answering our earlier question: position is all you need for token compression in the MLLM-based RES task.

| Method | refCOCO | refCOCO+ | refCOCOg | Avg. |
|---|---|---|---|---|
| random | 64.7 | 57.5 | 60.1 | 60.8 |
| L2 norm | 52.9 | 47.6 | 50.4 | 50.3 |
| sobel | 46.6 | 40.5 | 47.1 | 46.9 |
| canny | 50.0 | 43.7 | 44.2 | 43.7 |
| img clustering | 64.0 | 56.6 | 59.3 | 60.0 |
| emb clustering | 63.9 | 56.7 | 58.5 | 59.8 |
| feature FPS | 65.8 | 59.4 | 61.4 | 62.3 |
| PAYN (spatial FPS) | 69.2 | 61.8 | 64.2 | 65.2 |
| PAYN (checkerboard) | 70.4 | 63.0 | 65.2 | 66.4 |

*Table 2.* Performance of PAYN on Text4Seg$_{\text{LLaVA-1.5-7B}}$ under varying visual token counts. Avg. denotes the average cIoU across Referring Expression Segmentation datasets, and Rel. shows the percentage of performance retained at each token reduction level.

| Ratio | Method | refCOCO | | | refCOCO+ | | | refCOCOg | | Avg. | Rel. |
|---|---|---|---|---|---|---|---|---|---|---|---|
| | | val | testA | testB | val | testA | testB | val | test | | |
| *576 Tokens* (100%) | Vanilla | 79.3 | 81.9 | 76.2 | 72.1 | 77.6 | 66.1 | 72.1 | 73.9 | 74.9 | 100% |
| *384 Tokens* (↓ **33.3**%) | ToMe (ICLR23) | 8.6 | 8.7 | 8.5 | 7.4 | 7.5 | 7.1 | 7.3 | 7.7 | 7.9 | 10.5% |
| | VisionZip (CVPR25) | 24.1 | 21.9 | 26.4 | 22.9 | 21.1 | 23.2 | 24.3 | 24.3 | 23.5 | 31.4% |
| | Dart (EMNLP25) | 75.3 | 78.2 | 72.3 | 68.5 | 74.0 | 62.6 | 69.1 | 70.6 | 71.3 | 95.2% |
| | VisPruner (ICCV25) | 76.1 | 78.8 | 73.1 | **69.2** | 74.5 | 63.5 | 69.0 | 70.7 | 71.9 | 95.9% |
| | DivPrune (CVPR25) | 76.0 | 78.8 | 73.4 | 69.2 | 74.7 | 63.5 | 69.3 | 70.8 | 71.9 | 96.1% |
| | EVTP-IVS (WACV26) | 76.2 | 78.6 | 73.2 | 69.0 | 74.5 | 63.2 | 69.3 | **71.2** | 71.9 | 96.0% |
| | **PAYN** | **76.3** | **79.2** | **73.4** | 68.7 | **75.0** | **63.6** | **70.9** | 71.0 | **72.3** | **96.5%** |
| *288 Tokens* (↓ **50**%) | ToMe (ICLR23) | 14.2 | 14.8 | 13.1 | 12.7 | 12.3 | 10.8 | 13.1 | 13.5 | 13.1 | 17.4% |
| | VisionZip (CVPR25) | 18.1 | 17.5 | 17.9 | 16.9 | 16.7 | 16.3 | 18.0 | 17.0 | 17.3 | 23.1% |
| | Dart (EMNLP25) | 72.0 | 75.0 | 69.8 | 65.2 | 71.2 | 59.6 | 66.1 | 67.5 | 68.3 | 91.2% |
| | VisPruner (ICCV25) | 73.1 | 75.1 | 70.9 | 66.3 | 70.9 | 60.6 | 66.6 | 69.8 | 69.2 | 92.3% |
| | DivPrune (CVPR25) | 73.5 | 76.5 | 70.9 | 66.9 | 72.1 | 61.0 | 67.8 | 69.0 | 69.7 | 93.1% |
| | EVTP-IVS (WACV26) | 73.9 | 76.3 | 70.7 | 66.4 | 72.3 | 61.3 | 66.6 | 69.4 | 69.6 | 92.9% |
| | **PAYN** | **74.6** | **77.6** | **72.1** | **66.9** | **73.2** | **62.1** | **68.3** | **70.2** | **70.6** | **94.3%** |
| *192 Tokens* (↓ **66.7**%) | ToMe (ICLR23) | 14.2 | 14.0 | 13.2 | 12.0 | 12.6 | 11.4 | 13.2 | 12.4 | 12.9 | 17.2% |
| | VisionZip (CVPR25) | 11.8 | 12.2 | 10.8 | 10.6 | 10.8 | 9.0 | 7.3 | 10.0 | 10.3 | 13.7% |
| | Dart (EMNLP25) | 64.0 | 66.0 | 61.9 | 58.0 | 60.9 | 53.8 | 59.7 | 60.6 | 60.6 | 80.9% |
| | VisPruner (ICCV25) | 65.6 | 66.8 | 64.9 | 60.6 | 66.6 | 55.5 | 60.3 | 61.9 | 62.8 | 83.8% |
| | DivPrune (CVPR25) | 67.0 | 69.2 | 65.8 | 60.8 | 64.9 | 55.7 | 61.9 | 63.4 | 63.6 | 84.9% |
| | EVTP-IVS (WACV26) | 67.4 | 69.8 | 65.9 | 60.9 | 66.0 | 55.8 | 61.5 | 63.4 | 63.8 | 85.2% |
| | **PAYN** | **70.4** | **72.4** | **68.6** | **62.8** | **67.7** | **58.6** | **64.0** | **66.4** | **66.4** | **88.6%** |

As shown in Table 4, our position-based strategy consistently outperforms all alternative methods. Methods based on the L2 norm or edge detectors show inferior performance because they tend to select tokens concentrated in high-activation or boundary regions. Such selections fail to adequately cover all local neighboring regions, resulting in an imbalanced spatial distribution. In particular, the comparison between farthest point sampling in the feature space and in the spatial space highlights the dominant role of positional information, which even outweighs feature information for token selection. These results provide a clear answer to the question raised in *Chapter 3: **Yes, position is all you need for token compression in the MLLM-based RES task.***

**Effectiveness of Position Index Preserving** As shown in Table 5, we remove the preserved original positional indices in PAYN and replace them with continuous ones, resulting in a severe performance degradation. This finding is consistent with the results reported in Table 1, which show similar trends across other methods. These results demonstrate ***the effectiveness of the position index preserving module and further validate Insight 1***.

*Table 5.* Effectiveness of position index preserving.

| Method | refCOCO | refCOCO+ | refCOCOg | Avg. |
|---|---|---|---|---|
| PAYN w/o pos id | 11.1 | 9.6 | 9.5 | 10.2 |
| PAYN | 70.4 | 63.0 | 65.2 | 66.4 |

**Effectiveness of Position-Guided Token Selection** To validate our position-guided token selection strategy, which emphasizes retaining sufficient tokens within each local neighboring region, we compare it with the following token selection strategies that are also purely position-based:

(1) Center-biased sampling, which performs non-uniform sampling with higher density near the image center.
(2) Fixed-stride row/column sampling, which retains entire rows and columns at a fixed stride.
(3) Group-wise random sampling, which divides tokens into consecutive groups of equal size and randomly selects one token from each group.

As shown in Table 6, our token selection strategy achieves superior performance, which can be attributed to its ability to ensure broad and uniform spatial coverage and to better preserve local spatial structures. These results demonstrate ***the effectiveness of the position-guided token selection and further validate Insight 2***.

*Table 6.* Effectiveness of position-guided token selection.

| Method | refCOCO | refCOCO+ | refCOCOg | Avg. |
|---|---|---|---|---|
| center-biased | 59.9 | 52.5 | 55.8 | 56.1 |
| fixed-stride row | 67.7 | 59.5 | 62.4 | 63.3 |
| fixed-stride column | 65.5 | 58.1 | 61.5 | 61.7 |
| group-wise random | 68.4 | 60.0 | 63.8 | 64.1 |
| PAYN (spatial FPS) | 69.2 | 61.8 | 64.2 | 65.2 |
| PAYN (checkerboard) | 70.4 | 63.0 | 65.2 | 66.4 |

*Table 3.* Performance of PAYN on InstructSeg under varying visual token counts.

| Ratio | Method | refCOCO | | | refCOCO+ | | | refCOCOg | | Avg. | Rel. |
|---|---|---|---|---|---|---|---|---|---|---|---|
| | | val | testA | testB | val | testA | testB | val | test | | |
| *729 Tokens* (100%) | Vanilla | 85.1 | 85.8 | 83.7 | 81.1 | 84.1 | 78.1 | 79.6 | 80.3 | 81.9 | 100% |
| *145 Tokens* (↓ **80%**) | ToMe (ICLR23) | 76.2 | 78.0 | 76.6 | 64.6 | 69.2 | 60.2 | 66.5 | 67.3 | 69.8 | 85.3% |
| | VisionZip (CVPR25) | 70.7 | 72.0 | 70.2 | 55.0 | 58.9 | 52.4 | 56.0 | 56.9 | 61.5 | 75.1% |
| | Dart (EMNLP25) | 80.9 | 81.9 | 78.4 | 70.0 | 72.4 | 65.9 | 70.3 | 72.3 | 74.0 | 90.4% |
| | VisPruner (ICCV25) | 72.4 | 72.7 | 72.7 | 58.1 | 61.9 | 56.3 | 60.5 | 62.2 | 64.6 | 78.9% |
| | DivPrune (CVPR25) | 80.2 | 81.0 | 78.6 | 72.0 | 76.6 | 68.3 | 73.6 | 73.9 | 75.5 | 92.2% |
| | EVTP-IVS (WACV26) | 80.3 | 81.8 | 78.6 | 73.0 | 77.3 | 68.9 | 74.7 | 74.3 | 76.1 | 92.9% |
| | **PAYN** | **82.7** | **83.4** | **82.5** | **74.5** | **77.5** | **69.8** | **75.1** | **76.0** | **77.7** | **94.8%** |
| *72 Tokens* (↓ **90%**) | ToMe (ICLR23) | 74.6 | 76.0 | 74.4 | 60.2 | 64.5 | 56.9 | 62.3 | 64.0 | 66.6 | 81.3% |
| | VisionZip (CVPR25) | 70.6 | 71.0 | 69.7 | 52.1 | 56.0 | 49.8 | 54.7 | 55.0 | 59.9 | 73.1% |
| | Dart (EMNLP25) | 75.3 | 75.8 | 73.9 | 61.6 | 63.8 | 57.7 | 63.9 | 67.2 | 67.4 | 82.3% |
| | VisPruner (ICCV25) | 70.8 | 72.0 | 71.4 | 55.7 | 59.7 | 53.0 | 58.5 | 58.9 | 62.5 | 76.3% |
| | DivPrune (CVPR25) | 75.6 | 76.9 | 74.5 | 65.2 | **69.0** | 62.4 | 67.5 | 68.4 | 69.9 | 85.4% |
| | EVTP-IVS (WACV26) | 75.4 | 77.2 | 74.8 | 66.0 | 68.8 | 61.5 | 68.7 | 68.8 | 70.2 | 85.7% |
| | **PAYN** | **79.1** | **78.8** | **78.3** | **66.0** | 67.8 | **63.6** | **69.1** | **71.0** | **71.7** | **87.6%** |

**Efficiency Analysis**   As shown in Table 7, we evaluate the efficiency of PAYN on both the Text4Seg and InstructSeg baselines, reporting the total runtime, inference speedup, and TFLOPs. The results demonstrate that PAYN achieves an effective trade-off between model performance and computational efficiency. In addition, PAYN performs token selection solely based on positional information, without relying on token features or attention scores. The selection masks can even be precomputed and reused during inference, introducing no additional computational overhead.

*Table 7.* Efficiency analysis of PAYN on Text4Seg and InstructSeg. Metrics include total runtime (hour:min:sec), inference speedup, and TFLOPs on RefCOCO|TestA using a single A6000 GPU.

| Method | Token | Total time↓ | Δ ↑ | TFLOPs |
|---|---|---|---|---|
| Text4Seg | 576 | 11:18:42 | 1.0× | 8.94 |
| + VisPruner | 192 | 8:45:21 | 1.30× | 4.12 |
| + PAYN | 192 | 8:06:35 | 1.39× | 3.85 |
| InstructSeg | 729 | 28:12 | 1.0× | 2.38 |
| + VisPruner | 72 | 20:44 | 1.36× | 0.62 |
| + PAYN | 72 | 19:38 | 1.44× | 0.53 |

**Extension to Other MLLM Backbone**   As shown in Table 8, we further evaluate PAYN on the DeepSeekVL-based (Lu et al., 2024) Text4Seg baseline. PAYN consistently outperforms existing token compression methods, demonstrating its effectiveness and generalizability.

**Pre-Encoder vs. Post-Encoder Token Compression** Since PAYN is completely feature-agnostic and relies solely on positional information, we further investigate whether positional sampling can be directly applied to input image patches before feature extraction. As shown in Table 9, discarding patches before the vision encoder leads to a com-

*Table 8.* Comparison of PAYN and competing token compression methods on the Text4Seg baseline with DeepSeekVL-7B as the MLLM backbone, where 192 visual tokens are retained.

| Method | refCOCO | refCOCO+ | refCOCOg | Avg. |
|---|---|---|---|---|
| Vanilla | 78.4 | 71.9 | 74.4 | 75.0 |
| ToMe | 13.7 | 12.2 | 12.2 | 12.7 |
| VisionZip | 11.3 | 9.9 | 11.3 | 10.8 |
| Dart | 61.7 | 55.3 | 60.1 | 58.9 |
| VisPruner | 58.9 | 52.5 | 56.9 | 56.0 |
| DivPrune | 63.5 | 56.8 | 60.6 | 60.3 |
| EVTP-IVS | 64.8 | 57.1 | 60.6 | 60.9 |
| PAYN | 67.9 | 60.5 | 64.3 | 64.2 |

plete performance collapse. We attribute this to the fact that tokens in the vision encoder, especially in early layers, need to interact with neighboring tokens via self-attention to progressively aggregate local features into global representations; patches discarded prior to the vision encoder cannot participate in this process and their information is therefore permanently lost. In contrast, our method discards tokens after the vision encoder and before the LLM, where the retained tokens have already integrated contextual information from the discarded ones.

*Table 9.* Applying token compression prior to the vision encoder causes severe performance collapse.

| Method | refCOCO | refCOCO+ | refCOCOg | Avg. |
|---|---|---|---|---|
| pre-encoder | 10.9 | 9.4 | 9.6 | 9.9 |
| PAYN | 70.4 | 63.0 | 65.2 | 66.4 |

**Visualization Results**   As illustrated in Fig 6, we present two cases, where our method effectively preserves spatial structures, thereby achieving strong performance on dense prediction tasks such as RES.

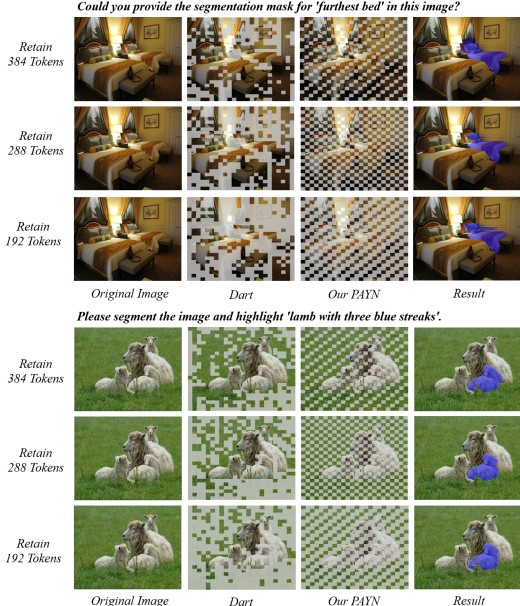

*Figure 6.* Visualization Results.

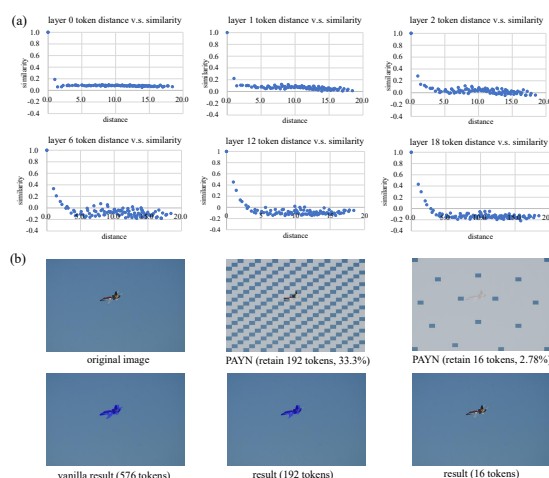

*Figure 7.* (a) Feature similarity of each token versus its distance to the foreground token in images with extremely small foregrounds. (b) PAYN can preserve local spatial structures and perform well under normal conditions, but may fail in extreme cases where objects are very small and only a few tokens are retained.

**Analysis on Extremely Small Objects** In extreme cases, the target object may occupy only a single image patch. Although PAYN is feature-agnostic and therefore cannot explicitly guarantee the retention of this token, object information can still spread to neighboring tokens through interactions in the vision encoder. Specifically, we select images with extremely small foregrounds and measure how each token's feature similarity varies with its distance to the foreground token, as shown in Fig. 8(a). We observe that, from shallow to deeper layers, feature similarity increases for nearby tokens while decreasing for distant ones. This indicates that even when a tiny object occupies only a single token, its information can still interact with neighboring patches. As shown in the left panel of Fig. 8(b), PAYN remains effective for extremely small objects while retaining only 192 tokens (33.3%).

## 6. Limitation and Future Direction

While effective in most settings, our method may exhibit limitations in extreme cases, such as when background regions contain little or no informative content while the number of retained tokens is extremely small, as shown in the right panel of Fig. 8(b). In such cases, PAYN may fail to preserve local spatial structures and therefore become suboptimal, and prioritizing regions with higher information content could be necessary. However, these cases are extremely uncommon and are not in conflict with our core perspective.

## 7. Conclusion

In this work, we observe that token compression for the MLLM-based referring expression segmentation (RES) task has been rarely explored, and existing methods developed for other vision-language tasks often suffer severe performance degradation when applied to RES. Through experiments and analysis, we find that positional information of visual tokens plays a substantially more critical role in RES than in other tasks. Motivated by this insight, we introduce PAYN, a plug-and-play, training-free token compression method that relies solely on positional information. Extensive experiments demonstrate that our method achieves improved inference efficiency while maintaining performance, validating our perspective that position is all you need for token compression in the MLLM-based RES task.

## Acknowledgments

This work is supported by the National Natural Science Foundation of China under grants 62206102; the National Key Research and Development Program of China under grant 2024YFC3307900; the National Natural Science Foundation of China under grants 62436003, 62376103 and 62302184; Major Science and Technology Project of Hubei Province under grant 2025BAB011 and 2024BAA008; Hubei Science and Technology Talent Service Project under grant 2024DJC078; and Ant Group through CCF-Ant Research Fund. The computation is completed in the HPC Platform of Huazhong University of Science and Technology.

## Impact Statement

This paper presents work whose goal is to advance the field of Machine Learning. There are many potential societal consequences of our work, none which we feel must be specifically highlighted here.

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

## A. Dataset Description

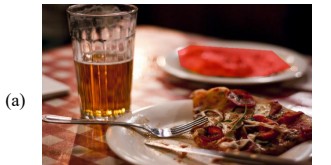
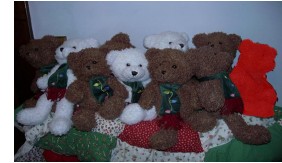

(a)

> Could you provide the segmentation mask for 'the food in the back right' in this image?

> Where is the very last bear on the right in this picture? Please respond with segmentation mask.

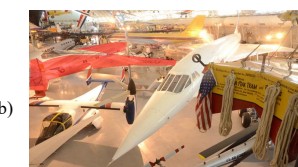
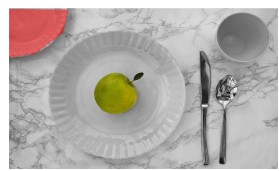

(b)

> Could you provide the segmentation mask for 'plane with numbers on wing' in this image?

> Please segment only the plate with no food in the image.

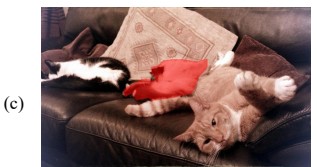
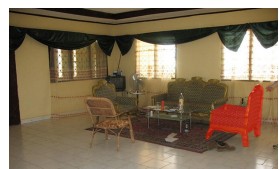

(c)

> Where is white and black cat laying on orange cat in this image? Please output segmentation mask.

> Could you provide the segmentation mask for a chair to the far right of the couch with gold trim in this image?

*Figure 8.* Samples from the (a) RefCOCO, (b) RefCOCO+, and (c) RefCOCOg datasets.

We evaluate our method on the RefCOCO, RefCOCO+ (Kazemzadeh et al., 2014), and RefCOCOg (Mao et al., 2016) benchmarks.
**RefCOCO** contains 19,994 images, 142,209 referring expressions, and 50,000 annotated objects. It supports referring expressions based on both spatial location and visual appearance attributes.
**RefCOCO+** includes 19,992 images, 141,564 expressions, and 49,856 annotated objects. Different from RefCOCO, it restricts the use of location-related descriptions and mainly focuses on appearance-based referring expressions.
**RefCOCOg** consists of 25,799 images with 95,010 referring expressions and 49,822 annotated objects. It is characterized by longer and more complex expressions without limitations on location references.

## B. Details of Baseline Methods

We employ two representative RES baselines: Text4Seg (Lan et al., 2025) and InstructSeg (Wei et al., 2025).

Text4Seg follows the text-as-mask paradigm, where the image is divided into a 16×16 grid of patches, and the MLLM assigns a semantic textual label to each patch in a row-wise

manner. For example, the segmentation mask for "dog laying down" can be represented as: <seg> others *16 \n dog laying down *1 | others *15 \n dog laying down *4 | others *12 \n others *16 . . . </seg>. To further improve the quality of the pixel-level semantic masks, SAM (Kirillov et al., 2023) is employed as a mask refinement module.

In contrast, InstructSeg is built upon an embedding-as-mask paradigm and supports segmentation for both images and videos. The framework mainly consists of an object-aware video perceiver (OVP), an MLLM, a visual encoder, a vision-guided multi-granularity text fusion module (VMTF), and a segmentation decoder. Specifically, the OVP compresses temporal and object-aware information from video frames into compact tokens, which are jointly processed with text tokens by the MLLM to generate mask embeddings and detailed text embeddings. Meanwhile, the visual encoder extracts semantic features from the input image or video frames. The generated embeddings and visual features are further fused by the VMTF module to enable comprehensive vision-language understanding, whose outputs are subsequently decoded by the segmentation decoder to generate segmentation masks and corresponding confidence scores.

Due to the differences in model architectures and structural complexity, we adopt different compression ratios for each RES baseline to achieve a better balance between segmentation performance and inference efficiency. Compared with Text4Seg, InstructSeg contains an additional vision encoder branch that helps alleviate the information loss caused by token compression. Moreover, InstructSeg follows an embedding-as-mask paradigm, where the MLLM predicts position-related embeddings rather than per-patch semantic classes as in Text4Seg, making it inherently less sensitive to suboptimal token compression. Therefore, we apply a higher compression ratio to the InstructSeg baseline.

## C. Hyperparameters of the Compared Methods

We use well-tuned configurations for all compared methods to ensure a fair comparison. Under the 192 tokens setting of the Text4Seg baseline, the hyperparameters of the comparison methods are summarized in Table 10. For InstructSeg, we adopt the same settings.

*Table 10.* Hyperparameters of the compared methods.

| Method | Hyperparameters |
|---|---|
| ToMe | layer=-2, token=192 |
| VisionZip | dominant=162, contextual=30 |
| Dart | pivot=48, duplicate=144 |
| VisPruner | important ratio=0.25, removal per iteration=8 |
| DivPrune | token=192 |
| EVTP-IVS | token=192 |

