# OpenReview forum: "Position Is All You Need: A Free Lunch Token Compression Strategy for MLLM-based Referring Expression Segmentation"
_ICML.cc/2026/Conference — ICML 2026 regular_

### Official Review · Reviewer_gtBS · 2026-02-25

**Soundness:** 3
**Presentation:** 3
**Significance:** 2
**Originality:** 3
**Overall Recommendation:** 4
**Confidence:** 3

**Summary:**

This paper investigates why standard visual token compression methods drastically degrade performance on MLLM-based Referring Expression Segmentation (RES), a dense prediction task. Through controlled analyses, the authors argue that preserving original positional indices and maintaining local spatial neighborhood structure are crucial for RES, unlike many sparse V+L tasks. Based on this, they propose PAYN, a training-free, plug-and-play token compression method that selects tokens purely by position (checkerboard or farthest point sampling) and preserves original position indices, achieving strong speedups with substantially better cIoU retention than prior compression baselines across RefCOCO/+/g on two RES frameworks (Text4Seg and InstructSeg).

**Compliance With Llm Reviewing Policy:**

Affirmed.

**Final Justification:**

The authors have resolved my concerns and I keep the score as weak accept.

**Key Questions For Authors:**

- LLaVA1.5 use absolute position encoding which is learnable. Are the experiment conducted on this?
- Is it possible to generalize into RoPE or Sin-based?
- Video-based Referring Expression Segmentation?

**Limitations:**

yes

**Strengths And Weaknesses:**

- **Strengths**
    - The method demonstrates improved results on standard benchmarks, particularly the RefCOCO series.
    - The paper provides a valuable observation that, unlike in image classification, preserving the spatial order and structure within positional encodings is critical for dense prediction tasks.

- **Weaknesses**
    - The proposed token compression strategy is primarily evaluated on specific baselines. It remains unclear whether this method generalizes effectively to other representative architectures, such as InstructSeg [1].
    - TYPO: L327: Table 4 rather than table 4.

[1] InstructSeg: Unifying Instructed Visual Segmentation with Multi-modal Large Language Models

---

> ### Author Rebuttal · Authors · 2026-03-31
>
> Thank you for your appreciation on our work.
>
> # 1. (W1&W2) Generalization to Other Architectures
>
> About Weakness 1, we have validated our method on **InstructSeg** in Table 3. As described in L302–308, we conduct experiments on: **Text4Seg under the text-as-mask paradigm (Table 2) and InstructSeg under the embedding-as-mask paradigm (Table 3)**, showing that our method generalizes across different architectures. The results on InstructSeg (Table 3) with 145 retained tokens can be summarized as follows:
>
> |Method|refcoco|refcoco+|refcocog|Avg.|
> |-|-|-|-|-|
> |Vanilla(729 tokens)|84.7|81.1|80.0|81.9|
> |ToMe|76.9|64.7|66.9|69.8|
> |VisionZip|71.0|55.4|56.5|61.5|
> |Dart|80.4|69.4|71.3|74.0|
> |VisPruner|72.6|58.8|61.4|64.6|
> |DivPrune|79.9|72.3|73.8|75.5|
> |EVTP-IVS|80.2|73.1|74.5|76.1|
> |**PAYN**|**82.8**|**73.9**|**75.5**|**77.7**|
>
> About Weakness 2, thank you for pointing out the typo, we will correct it in the paper.
> # 2. (Q1&Q2) Position Encoding
>
> There are **two position encodings in MLLMs: one in the vision encoder and one in the LLM**. In most MLLMs, **the vision encoder uses learned absolute position embeddings**, and the generated visual tokens are then fed together with text tokens into the **LLM, whose internal position encoding is RoPE**. Our method is validated on **LLaVA-1.5** (Text4Seg baseline) and **Mipha** (InstructSeg baseline), whose position encodings are consistent with the above. Our approach preserves token position indices **at the LLM input stage, operating on RoPE and independent of the vision encoder’s internal position encoding**.
>
> We further apply our method to **DeepseekVL** (Text4Seg baseline), which, like most MLLMs, uses RoPE in the LLM. Results for 192 retained tokens:
>
> |Method|refcoco val|refcoco testA|refcoco testB|Avg.|
> |-|-|-|-|-|
> |Vanilla(576 tokens)|76.3|72.5|74.7|74.5|
> |ToMe|13.7|12.2|12.2|12.7|
> |VisionZip|11.3|9.9|9.9|10.4|
> |Dart|61.7|55.3|60.1|58.9|
> |VisPruner|62.4|56.2|60.7|59.6|
> |DivPrune|63.5|56.8|60.6|60.3|
> |EVTP-IVS|64.8|57.1|60.6|60.9|
> |**PAYN**|**67.9**|**60.5**|**64.3**|**64.2**|
>
> The results show the effectiveness and generalization of our approach.
>
> # 3. (Q3) Evaluation on Video-based Referring Expression Segmentation
>
> We evaluate our method on video-level tasks:
>
> - **Referring Video Object Segmentation (RVOS)**: focusing on referring expressions. Dataset: Ref-DAVIS17, metric: J&F. The vanilla baseline (InstructSeg, 729 tokens) achieves a J&F of 71.1.
> - **Reasoning Video Object Segmentation (ReVOS)**: focusing on complex implicit reasoning. Dataset: ReVOS, metric: J&F. The vanilla baseline (InstructSeg, 729 tokens) achieves a J&F of 51.9.
>
> |Method|145 tokens (↓80%) RVOS J&F|145 tokens (↓80%) ReVOS J&F|72 tokens (↓90%) RVOS J&F|72 tokens (↓90%) ReVOS J&F|
> |-|-|-|-|-|
> |ToMe|62.7|41.6|62.1|42.5|
> |VisionZip|58.3|37.6|56.7|38.6|
> |Dart|67.3|46.3|64.2|44.0|
> |VisPruner|60.0|40.9|60.8|41.7|
> |DivPrune|63.9|45.9|63.4|47.7|
> |EVTP-IVS|64.7|46.4|64.5|48.2|
> |**PAYN**|**68.2**|**46.9** |**65.4**|**48.4**|
>
> The results demonstrate the effectiveness of our method.

---

> > ### Author Rebuttal · Reviewer_gtBS · 2026-04-05
> >
> > The authors have fully addressed my concerns.

---

> > > ### Author Response · Authors · 2026-04-05
> > >
> > > Thank you again for your recognition of our work, we will carefully incorporate your suggestions in the revised version.

---

### Official Review · Reviewer_jziX · 2026-03-12

**Soundness:** 3
**Presentation:** 3
**Significance:** 3
**Originality:** 3
**Overall Recommendation:** 4
**Confidence:** 4

**Summary:**

This paper studies token compression for Multimodal Large Language Model (MLLM)-based Referring Expression Segmentation (RES). While token compression methods have been widely explored to reduce the quadratic cost of attention in vision-language models, the authors observe that existing approaches lead to severe performance degradation when applied to RES tasks.

Through empirical analysis, the paper finds that preserving positional information and local spatial neighborhood structures is critical for RES performance. Based on this observation, the authors propose PAYN, a training-free token compression strategy that selects tokens solely based on their spatial positions. The method preserves the original positional indices while retaining tokens that are evenly distributed across local regions.

The proposed approach is plug-and-play and can be applied without retraining existing MLLM-based RES models. Experiments on multiple RES benchmarks demonstrate that PAYN significantly outperforms existing token compression strategies under the same compression ratios.

**Compliance With Llm Reviewing Policy:**

Affirmed.

**Final Justification:**

My main concerns are addressed in the rebuttal.

**Key Questions For Authors:**

1. The proposed method uses a fixed spatial distribution of tokens. Have the authors considered adaptive variants that preserve positional coverage while still allowing token importance weighting?

2. What is the actual inference speedup or FLOP reduction obtained when applying PAYN in practice?

3. Can the method be extended to video-based referring segmentation or dense prediction tasks beyond RES?

**Limitations:**

Yes.

**Strengths And Weaknesses:**

Strength
1. Clear empirical insight.
The paper identifies an important and somewhat overlooked issue: conventional token compression strategies that work well for general vision-language tasks fail on RES tasks. The observation that positional information is critical for segmentation reasoning is intuitive but useful and well-supported by experiments.

2. Simple and practical method.
PAYN is training-free and architecture-agnostic, making it easy to integrate into existing pipelines. This practical design makes the contribution appealing for real-world deployment where retraining large MLLMs is expensive.

3. Strong efficiency-performance tradeoff.
The method achieves competitive or superior performance compared to prior token compression methods while reducing computational overhead. The efficiency gains appear meaningful given the heavy cost of MLLM inference.

4. Well-motivated design.
The proposed method directly follows from the empirical findings regarding positional structure preservation. The connection between the analysis and the final method is clear.

Weaknesses
1. Limited novelty of the compression strategy.
While the empirical analysis regarding positional information is interesting, the final method itself is relatively simple and resembles spatial sampling strategies that preserve uniform spatial coverage.

2. Limited theoretical explanation.
The paper empirically shows that positional information is critical for RES, but the underlying reasons for why RES models are particularly sensitive to token position are not deeply analyzed.

3. Fixed compression strategy.
The proposed approach uses a deterministic spatial selection strategy. It would be interesting to explore whether adaptive variants that preserve spatial coverage while incorporating token importance could further improve performance.

---

> ### Author Rebuttal · Authors · 2026-03-31
>
> Thank you for your thorough review and constructive feedback. Below are our detailed responses to your concerns:
>
> # 1. (W1) Simple but Practical
>
> Although our method is simple, we have conducted extensive comparisons from the perspective of different baselines (Table 2, 3), token selection strategies (Table 2–4), and position-based strategies (Table 6) to validate its effectiveness. A simplified summary is provided below:
>
> |Method|Avg.|
> |-|-|
> |ToMe|12.9|
> |VisionZip|10.3|
> |Dart|60.6|
> |VisPruner|62.8|
> |DivPrune|63.6|
> |EVTP-IVS|63.8|
> |Random|60.8|
> |L2 norm|50.3|
> |Sobel|46.9|
> |Canny gaussian blur|43.7|
> |Image kmeans|60.0|
> |Patch embedding kmeans|59.8|
> |Feature fps|62.3|
> |Fixed stride line/column|63.3/61.7|
> |Group wise random|64.1|
> |**Ours(spatial fps)**|**65.2**|
> |**Ours(checkerboard)**|**66.4**|
>
> Our method’s advantages lie in:
>
> - **Novelty**: It focuses on token compression in the **underexplored RES task** and is built upon the insight that position is the most critical factor, thoroughly validated by our experiments and analysis. Although some methods attempt to select diverse tokens (e.g., DivPrune), they do so in **feature space rather than considering position or spatial coverage**.
> - **Performance**: It **outperforms** both non-position-based token selection methods (e.g., VisionZip) and other position-based strategies (e.g., fixed-stride sampling).
> - **Efficiency**: It is a **free-lunch** solution and introduces no additional computational overhead.
>
> Further evidence is provided in our response 3 and 5.
>
> # 2. (W2) Theoretical Explanation
>
> In Section 3.3, we conducted experiments and analysis on sparse and dense prediction tasks. The sensitivity of RES to the preservation of positional information can be interpreted from a **sharpness** [1] perspective in the input space:
>
> $$
> \text{Sharpness} _ {\text{input}} = \max _ {\|\delta\| \le \rho} \big( \mathcal{L}(X + \delta) - \mathcal{L}(X) \big)
> \approx \frac{1}{2} \, \delta^\top H _ X \, \delta
> $$
>
> Here, δ represents small perturbations applied to the input (e.g. TPS deformation), and H_X is the Hessian capturing the second-order sensitivity of the loss.
>
> Prior work [2] shows that **dense prediction tasks rely heavily on high-frequency spatial components. High-frequency components naturally have larger second-order derivatives, resulting in larger Hessian norms and sharper loss landscapes.** Consequently, under the same perturbation δ, the change in loss L(X+δ)−L(X) is greater in dense prediction tasks like RES, leading to more significant performance degradation compared to sparse prediction tasks.
>
> Other token selection strategies are more spatially concentrated than our method; thus, when selected tokens are unreliable (e.g., due to hallucination or attention sink), the errors are difficult to compensate since neighboring regions may not be covered. **In contrast, our spatially averagely distributed strategy allows unselected tokens to be better represented by nearby retained ones, improving robustness for RES, which exhibits high sharpness and sensitivity.**
>
> [1] Sharpness-Aware Minimization for Efficiently Improving Generalization
>
> [2] Understanding the Robustness of Vision Transformers
>
> # 3. (W3&Q1) Fixed Strategy vs. Adaptive Variants
>
> We retain 192 tokens (33.3%) and compare our method with several importance-aware alternatives:
>
> |Method|refcoco val|refcoco testA|refcoco testB|Avg.|
> |-|-|-|-|-|
> |**Ours**|**70.1**|**72.2**|**68.9**|**70.4**|
> |Attn only|56.0|57.4|56.5|56.4|
> |Adaptive by attention|65.4|67.7|64.8|65.9|
> |Adaptive by attention-based window|61.8|63.4|62.2|62.3|
>
> - Adaptive by attention: select a subset of tokens by attention and the rest via our positional sampling
> - Adaptive by attention-based window: select tokens via our positional sampling within high-attention windows
>
> **Our method consistently outperforms them, demonstrating its effectiveness**.
>
> # 4. (Q2) Actual Inference Speedup
>
> We reported the TFLOP reduction in L408–412. The actual inference speedup in practice is as follows:
>
> |Method|Token|Total Time|∆|TFLOPs|
> |-|-|-|-|-|
> |Text4Seg|576|11:18:42|1.0×|8.94|
> |+VisPruner|192|8:45:21|1.30×|4.12|
> |**+PAYN**|**192**|**8:06:35**|**1.39×**|**3.85**|
> |InstructSeg|729|28:12|1.0×|2.38|
> |+VisPruner|192|20:44|1.36×|0.62|
> |**+PAYN**|**72** |**19:38**|**1.44×**|**0.53**|
>
> # 5. (Q3) Extension to Tasks Beyond RES
>
> We further evaluate our method on **Reasoning Segmentation (ReasonSeg), Referring Video Object Segmentation (RVOS) and Reasoning Video Object Segmentation (ReVOS)**, reporting results with 145 tokens. Due to space limitation, please refer to reviewer 94CR's reply2 for more details.
>
> |Method|ReasonSeg cIoU|RVOS J&F|ReVOS J&F|
> |-|-|-|-|
> |Vanilla(729 tokens)|65.2|71.1|51.9|
> |ToMe|52.1|62.7|41.6|
> |VisionZip|49.5|58.3|37.6|
> |Dart|60.4|67.3|46.3|
> |VisPruner|52.4|60.0|40.9|
> |DivPrune|53.0|63.9|45.9|
> |EVTP-IVS|56.1|64.7|46.4|
> |**PAYN**|**60.4**|**68.2**|**46.9**|

---

> > ### Author Rebuttal · Reviewer_jziX · 2026-04-03
> >
> > Thank the authors for the explanation, I will change my score to 4.

---

> > > ### Author Response · Authors · 2026-04-03
> > >
> > > Thank you very much for agreeing to raise the score. We will carefully incorporate your suggestions in the revised version.
> > >
> > > Regarding the score, we would greatly appreciate it if you could kindly update it in the system when convenient.
> > >
> > > Thank you again for your time and support.

---

### Official Review · Reviewer_u72T · 2026-03-12

**Soundness:** 2
**Presentation:** 3
**Significance:** 3
**Originality:** 3
**Overall Recommendation:** 4
**Confidence:** 5

**Summary:**

This paper investigates the performance degradation of existing visual token compression methods when applied to MLLM-based Referring Expression Segmentation (RES). The authors observe that dense prediction tasks are highly sensitive to local spatial structures. Based on this, they propose a straightforward, training-free token compression strategy named PAYN. Instead of relying on semantic similarity or attention, PAYN uniformly samples tokens in the spatial domain (e.g., Checkerboard or Farthest Point Sampling) while explicitly preserving their original positional indices. While the observation is interesting, the paper falls short in terms of methodological novelty, comprehensiveness of baselines, and granular experimental analysis.

**Compliance With Llm Reviewing Policy:**

Affirmed.

**Final Justification:**

The rebuttal has resolved my major concerns. The authors should ensure that the discussion on feature similarity across layers and the acknowledged limitations regarding extreme sparsity are explicitly incorporated into the final version of the manuscript, as they significantly strengthen the paper's theoretical grounding.

I will adjust my score to reflect these clarifications.

**Key Questions For Authors:**

Refer to the weaknesses.

**Limitations:**

Yes

**Strengths And Weaknesses:**

Strengths:

- The PAYN method is straightforward and training-free. Since it relies purely on positional indices rather than complex feature clustering or attention sorting, the selection masks can be precomputed. This keeps the additional inference overhead minimal.

- The method demonstrates robust performance retention at high compression rates. For example, on the Text4Seg baseline, pruning visual tokens from 576 to 192 (about 66.7% reduction) still yields an average cIoU of 66.4, which significantly outperforms competitive baselines like DART.

Weaknesses:
- Because it uses purely positional sampling and ignores actual image content, PAYN risks dropping tokens that capture critical, fine-grained details. If a prompt asks to segment a tiny object, uniform sampling might just miss it. The authors briefly mention extreme cases, but the paper really needs a performance breakdown by object size (Small/Medium/Large) to actually quantify this trade-off.

- The claims in Section 3.3 regarding "Dense Prediction Tasks" feel a bit broad given the current experiments. To properly support this, please include results on at least one other standard dense task using MLLMs (e.g., general semantic segmentation, instance segmentation, or object detection).

- Please provide the exact experimental settings and hyperparameters used for the comparison methods (especially VisionZip) across both the Text4Seg and InstructSeg environments. Could you please explain the discrepancy where VisionZip completely collapses in Table 2 but performs reasonably well in Table 3？

- If PAYN is completely feature-agnostic and relies only on positional indices, it raises the question of whether extracting full-resolution features is even necessary. The authors should add a baseline where this positional sampling is applied directly to the input image patches prior to the vision encoder.

---

> ### Author Rebuttal · Authors · 2026-03-31
>
> Thank you for your thorough review and constructive feedback. Below are our detailed responses to your concerns:
>
> # 1. Performance Across Object Sizes
>
> We divide the RefCOCO dataset into five categories (huge, large, medium, small, tiny) by the ground-truth mask area ratio for fine-grained evaluation. Retaining 192 tokens (33.3%) on Text4Seg baseline, we compare our method with importance-aware strategies:
>
> |Method|H|L|M|S|T|overall|
> |-|-|-|-|-|-|-|
> |**Ours**|**75.3**|**74.0**|**72.2**|**67.8**|**64.5**|**70.4**|
> |Attn only|54.4|49.2|45.8|40.8|36.2|56.4|
> |Half attn half fixed|63.8|55.4|49.9|43.1|35.1|65.9|
> |Fixed in high-attn windows|60.3|54.6|50.8|45.6|40.5|62.3|
>
> We find that **even for tiny objects (foreground <3.33% of the image), our method still outperforms others**. This is because, although our method may not precisely locate every tiny object, it **(1) covers neighboring regions, allowing unselected tokens to be represented by nearby retained ones, and (2) preserves local structures**. In contrast, when other methods miss the target (e.g., due to hallucination or attention sink), they **(1) lack neighboring tokens to compensate for the lost target information and (2) fail to preserve local structures.**
>
> We further reduce the number of retained tokens to 48 (8.33%), and observe that the cIoU on tiny objects for our method (15.2) is slightly lower than Attn only (19.4); but **such aggressive sacrifices of performance for efficiency rarely occur**. Data splits and extreme-case visualizations are available at: https://anonymous.4open.science/r/vis-1450/rebuttal_pic2.png.
>
> # 2. Evaluation on Additional Dense Prediction Tasks
>
> We further evaluate our method on **Reasoning Segmentation (ReasonSeg), Referring Video Object Segmentation (RVOS) and Reasoning Video Object Segmentation (ReVOS)**, reporting results with 145 tokens. Due to space limitation, please refer to reviewer 94CR's reply2 for more details.
>
> |Method|ReasonSeg cIoU|RVOS J&F|ReVOS J&F|
> |-|-|-|-|
> |Vanilla(729 tokens)|65.2|71.1|51.9|
> |ToMe|52.1|62.7|41.6|
> |VisionZip|49.5|58.3|37.6|
> |Dart|60.4|67.3|46.3|
> |VisPruner|52.4|60.0|40.9|
> |DivPrune|53.0|63.9|45.9|
> |EVTP-IVS|56.1|64.7|46.4|
> |**PAYN**|**60.4**|**68.2**|**46.9**|
>
> # 3. Experimental Details and Analysis of VisionZip Performance
>
> **Hyperparameters**
>
> Our experiments are conducted on NVIDIA RTX A6000 GPU. The Text4Seg hyperparameters (192 tokens) are:
>
> ToMe: layer=−2, token=192
>
> VisionZip: dominant=162, contextual=30
>
> DART: pivot=48, duplicate=144
>
> VisPruner: important ratio=0.25, removal per iteration=8
>
> DivPrune/EVTP-IVS: token=192
>
> We use well-tuned configurations for all compared methods to ensure a fair comparison. For InstructSeg, we adopt the same settings, and our reproduced results are consistent with those reported in EVTP-IVS (within ±2 cIoU). Therefore, we report their results for consistency.
>
> **Why VisionZip collapses on Text4Seg**
>
> VisionZip performs contextual token merging, making position IDs impossible to preserve and thus losing spatial information critical for RES. As shown in Table 1 and L185–192, retaining only dominant tokens with position IDs greatly improves performance.
>
> |Method|refcoco val|refcoco testA|refcoco testB|Avg.|
> |-|-|-|-|-|
> |dom162, con30|11.8|12.2|10.8|11.6|
> |dom192|11.1|10.8|9.9|10.6|
> |con192|9.8|10.0|9.5|9.9|
> |**dom192+our Pos-id-preserving**|**56.2**|**57.0**|**56.3**|**56.5**|
>
> **Why VisionZip performs reasonably well on InstructSeg**
>
> VisionZip still performs worse than other methods on InstructSeg but does not collapse. This is because InstructSeg has an **extra vision encoder branch** that mitigates information loss, leading to a higher performance retention than Text4Seg across various compression methods (e.g., ToMe, DART, DivPrune, EVTP-IVS) even under more aggressive compression ratios (L276–281). Besides, it adopts an **embedding-as-mask paradigm**, where the MLLM predicts position-related embeddings rather than per-patch classes as in Text4Seg, making it less sensitive to suboptimal token compression.
>
> # 4. Ablation on Pre-Encoder Positional Sampling
>
> We apply positional sampling prior to the vision encoder:
>
> |Method|refcoco|refcoco+|refcocog|Avg.|
> |-|-|-|-|-|
> |**Ours**|**70.4**|**63.0**|**65.2**|**66.3**|
> |Pre-encoder|10.9|9.4|9.6|9.9|
>
> We observe a total performance collapse when patches are discarded before the vision encoder. This is because tokens in the vision encoder, especially in early layers, need to **interact with neighboring tokens via self-attention to progressively aggregate local features into global representations; patches discarded prior to the vision encoder cannot participate in this process and their information is permanently lost**. In contrast, our method discards tokens after the vision encoder and before the LLM, where the **retained tokens have already integrated information from discarded ones**.

---

> > ### Author Rebuttal · Reviewer_u72T · 2026-04-02
> >
> > While I appreciate the authors' detailed explanation and the new evaluation on tiny objects (<3.33%), the argument regarding 'attention sinks' seems somewhat asymmetrical.
> >
> > If PAYN relies strictly on uniform positional sampling without any semantic guidance, it will inevitably sample these exact 'attention sink' patches as well, thereby wasting valuable token budget on uninformative regions. Furthermore, relying on adjacent tokens to compensate for discarded tiny objects assumes that sufficient target features have already leaked into neighboring patches. However, if the tiny object occupies only a single patch that happens to be discarded, a feature-agnostic approach fundamentally lacks a mechanism to guarantee its retention.
> >
> > Could the authors clarify why preserving local structures via rigid positional sampling is definitively better than an improved semantic-aware selection that actively filters out sinks?

---

> > > ### Author Response · Authors · 2026-04-05
> > >
> > > Many thanks for your careful and thoughtful comments! Here are our responses:
> > >
> > > # The Rationality of Our Method
> > >
> > > **(1) About attention sink.** We would like to clarify that as attention sinks are not inherently beneficial or harmful [1], our method neither favors nor avoids sampling them. Instead, our point is that **token selection strategies built upon attention are not well-suited for RES tasks**: such methods tend to focus **solely** on salient regions, weakening the sampling of tokens in other regions, therefore harming the preservation of spatial structure across the whole image. As discussed in L240–242 (Insight 2), RES task relies more on the preservation of local spatial structures. In other words, the most important thing for this task is to keep the spatial structures, while it may not be so important to sample sink tokens or not.
> > >
> > > **(2) About tiny objects.** We select images with extremely small foregrounds and measure how each token’s feature similarity varies with its distance to the foreground token, as shown in https://anonymous.4open.science/r/vis-5587/rebuttal_pic3.png. **From early to deeper layers, similarity gradually increases for nearby tokens and decreases for distant ones, suggesting that even when a tiny object occupies only a single token, its information can still interact with neighboring patches.** Thus even if the target token is discarded, its information can still be partially recovered by neighboring tokens. Therefore, across experiments with varying object sizes, our method consistently achieves superior performance on this task.
> > >
> > > [1] See What You Are Told: Visual Attention Sink in Large Multimodal Models
> > >
> > > # The Optimality of Our Method
> > >
> > > We acknowledge that our method is **not optimal in all scenarios**. However, given that token compression for the dense prediction tasks is still underexplored, our solution is the most advanced one across several representative tasks so far.
> > >
> > > As discussed in Section 6 (Limitations and Future Directions) and mentioned in our previous response, since the perception of tiny objects relies on the sampled neighboring tokens, if objects are extremely small and the number of retained tokens is very limited, the sampled tokens would be sparse and may be far from the tiny object, and therefore our approach would be limited. In such extreme cases, more precise, object-aware token selection (while preserving spatial local structures across the image) could be preferable. However, since **(1) these cases are very uncommon, (2) our method can already provide better results in widely-adopted settings, and (3) token compression for RES task remains largely unexplored**, our approach, **as a foundational work, is the first to verify the importance of spatial local structures in this task, by designing a seemingly counter-intuitive method to solely rely on position information**. Acknowledging that our method is not optimal in all scenarios, as a pioneering work in this task, we therefore leave the exploration of specialized strategies for extreme cases to future work.

---

### Official Review · Reviewer_94CR · 2026-03-12

**Soundness:** 3
**Presentation:** 3
**Significance:** 3
**Originality:** 3
**Overall Recommendation:** 5
**Confidence:** 4

**Summary:**

The paper shows that typical compression methods for vision tokens come with significant performance losses. The paper proposes an explanation: the importance of visual token position information. The paper conducts numerous experiments to demonstrate this hypothesis and uses this observation to propose PAWN, a simple and efficient compression method. PAWN outperforms existing token compression methods on several benchmarks and models.

**Compliance With Llm Reviewing Policy:**

Affirmed.

**Final Justification:**

I keep my score. The rebuttal answered the few questions I had.

**Key Questions For Authors:**

- Do you know why the Checkerboard works better than spatial FPS? Please also specify which strategy was used in the captions of Table 2 and 3.
- Could you evaluate the performance of VLMs with PAWN for tasks other than RES? Could your method also be effective for other tasks?

**Limitations:**

yes

**Strengths And Weaknesses:**

**Soundness:**

The scientific approach is rigorous. The idea that information about the position of visual tokens is very important is demonstrated through numerous experiments and ablations, notably Table 1. Figure 4 shows that this information is particularly important for the task of Referring Expression Segmentation. The proposed method is therefore well-motivated, and the experiments in Table 2 and 3 show that PAWN performs significantly better than existing methods.

**Presentation:**

The paper is very well written and the approach is very well explained. Although they are cited in the paper, I think it would be interesting to introduce the Dart, VisPruner, PruMerge, and VisionZip methods with their name in the Related Work section (their names only appear in the Tables).

**Significance:**

This paper addresses an important issue, namely the pruning of visual tokens in the context of referential expression segmentation.

**Originality:**

The paper presents new insights using a new method.

---

> ### Author Rebuttal · Authors · 2026-03-31
>
> Thank you for your appreciation on our work.
>
> # 1. Checkerboard vs. Spatial FPS Performance
>
> Spatial FPS is a greedy farthest-point sampling strategy that maximizes pairwise distances, resulting in a **locally optimal but not globally uniform** partition. In contrast, Checkerboard performs regular and globally uniform sampling, forming a stable grid-like spatial coverage. Therefore Checkerboard slightly outperforms Spatial FPS. As shown in the anonymous GitHub link (https://anonymous.4open.science/r/vis-1450/rebuttal_pic1.png), we provide visualizations of the retained tokens for both methods.
>
> The results in Table 2 and 3 are based on the Checkerboard strategy. Thank you for pointing it out, we will specify this in the paper.
>
> # 2. Effectiveness on Tasks Beyond RES
>
> We further evaluate our method on:
>
> **Image task:**
>
> - **Reasoning Segmentation (ReasonSeg)**: image-level, focusing on complex implicit reasoning. Dataset: ReasonSeg, metric: gIoU, cIoU. The vanilla baseline (InstructSeg, 729 tokens) achieves a gIoU of 61.9 and a cIoU of 65.2.
>
> |Method|145 tokens (↓80%) gIoU|145 tokens (↓80%) cIoU|72 tokens (↓90%) gIoU|72 tokens (↓90%) cIoU|
> |-|-|-|-|-|
> |ToMe|48.9|52.1|48.0|53.6|
> |VisionZip|46.3|49.5|42.6|47.7|
> |Dart|54.1|60.4|53.2|54.3|
> |VisPruner|49.0|52.4|44.8|49.3|
> |DivPrune|52.3|53.0|50.9|52.6|
> |EVTP-IVS|54.3|56.1|51.3|55.6|
> |**PAYN**|**57.7**|**60.4**|**55.1**|**58.9**|
>
> **Video task:**
>
> - **Referring Video Object Segmentation (RVOS)**: video-level, focusing on referring expressions. Dataset: Ref-DAVIS17, metric: J&F. The vanilla baseline (InstructSeg, 729 tokens) achieves a J&F of 71.1.
> - **Reasoning Video Object Segmentation (ReVOS)**: video-level, focusing on complex implicit reasoning. Dataset: ReVOS, metric: J&F. The vanilla baseline (InstructSeg, 729 tokens) achieves a J&F of 51.9.
>
> |Method|145 tokens (↓80%) RVOS J&F|145 tokens (↓80%) ReVOS J&F|72 tokens (↓90%) RVOS J&F|72 tokens (↓90%) ReVOS J&F|
> |-|-|-|-|-|
> |ToMe|62.7|41.6|62.1|42.5|
> |VisionZip|58.3|37.6|56.7|38.6|
> |Dart|67.3|46.3|64.2|44.0|
> |VisPruner|60.0|40.9|60.8|41.7|
> |DivPrune|63.9|45.9|63.4|47.7|
> |EVTP-IVS|64.7|46.4|64.5|48.2|
> |**PAYN**|**68.2**|**46.9** |**65.4**|**48.4**|
>
> The results demonstrate the effectiveness of our method.
>
> # 3. Adding Method Names to Related Work
>
> Thank you for your suggestion. We will add the names of these methods to the Related Work section to further improve readability.

---

> > ### Author Rebuttal · Reviewer_94CR · 2026-04-03
> >
> > Thank you for your detailed response. I will keep my rating (accept).

---

> > > ### Author Response · Authors · 2026-04-03
> > >
> > > Thank you again for your recognition of our work, we will carefully incorporate your suggestions in the revised version.

---

### Decision · Program_Chairs · 2026-04-30

**Decision:**

Accept (regular)

**Comment:**

In this paper, the main concerns were limited novelty, evaluation breadth, and analysis depth. In the rebuttal, the authors provided additional results on object-size splits, other dense/video tasks, architectural generalization, runtime, and baseline details, which addressed most of these concerns. Overall, reviewers acknowledged that most concerns were resolved and were broadly satisfied after the rebuttal. Given the simplicity, training-free nature, and effectiveness of the proposed method, along with its solid empirical support and practical value, the AC recommends acceptance.